# Tobacco Use as a Health Disparity: What Can Pediatric Clinicians Do?

**DOI:** 10.3390/children6020031

**Published:** 2019-02-20

**Authors:** Jyothi Nagraj Marbin, Valerie Gribben

**Affiliations:** Department of Pediatrics, University of California, San Francisco General Hospital, San Francisco, CA 94110, USA; valerie.gribben@ucsf.edu

**Keywords:** health disparities, e-cigarettes, tobacco use, structural barriers, pediatricians, pediatrics

## Abstract

Tobacco use is a global health crisis, and has a tremendous and negative impact on health and wellbeing. Tobacco use disproportionately affects members of vulnerable populations, and by acting on multiple socioecological levels, serves to perpetuate and reinforce cycles of poverty. Members of the pediatric medical community can play a key role in interrupting cycles of tobacco use. Providers can serve as powerful allies to vulnerable communities by treating tobacco use in caregivers, counseling youth against using tobacco products, protecting children from the impact of secondhand smoke exposure, and advocating for economic, social, and health policies to disrupt intergenerational smoking.

## 1. Tobacco Use as a Health Disparity

Tobacco use remains the major cause of annual preventable death in the United States and globally [1]. Although rates of combustible tobacco cigarette smoking among US adults have fallen from over 40% in the 1960s to 14% in 2017 [2], smoking and secondhand smoke exposure (SHSE) persist as health disparities for people living in poverty and for other vulnerable groups. According to the Centers for Disease Control (CDC), “health disparities negatively affect groups of people who have systematically experienced greater social or economic obstacles to health” [3]; this is certainly true for tobacco use. More than 25% of people living below the federal poverty line are smokers, which is nearly double the rate of those living above the poverty line [4]. CDC data from 2016 (see Table 1) highlight the persistent differences in smoking rates. Smoking rates remain higher in people who have general education development certificates (GEDs) versus graduate degrees, those who have experienced psychological distress, and those living below the poverty line [5]. Those who are uninsured (27.9%) or on Medicaid (29.1%) smoke more often than those who are privately insured (12.9%) [6]. Seventy-five percent of homeless adults use tobacco [4]. Additionally, tobacco use is higher in other marginalized groups, including people who have mental illness, substance users, and those with a physical disability [7]. Young adults who are aging out of foster care have a daily smoking rate (32%) that is almost four times that of the general population of young adults [8].

This pattern persists internationally: Across the United States, Canada, New Zealand, and Australia, smoking rates range from 16% to 24%, but are dramatically higher in the homeless population (68%–89%), those with mental illness (30%–62%), and substance abusers (56%–93%) [9].

In the analysis of results from the National Health Interview Survey from 2013 [10], similar trends are seen among specific parent populations and are detailed in Table 2. Single parents were more likely to be smokers than parents in two-parent families. There were also differences seen among subsets of single parents, such as socioeconomic status, race, and educational attainment.

Why are marginalized groups using combustible tobacco so heavily? The National Cancer Institute uses a socioecological model to describe the myriad factors perpetuating tobacco use as a health disparity: these include individual factors such as psychological disorders and stress; interpersonal factors including racism, discrimination, and social relationships; community and neighborhood factors; and finally, societal factors like socioeconomic status (SES) [11]. We know that there is a strong pattern of intergenerational tobacco use; children whose parents smoke are much more likely to become smokers themselves, thus perpetuating generations of tobacco use [12]. There is evidence to support genetic predisposition to nicotine addiction, which can further perpetuate intergenerational nicotine addiction [13].

Societal-level factors, in the form of marketing and communications, perpetuate tobacco use. For years, tobacco companies have strategically targeted vulnerable populations [4]. Marginalized populations are exposed to a higher concentration of tobacco advertisements and retailers. The strategies used include handing out cigarettes in low-income housing and giving out tobacco coupons with food stamps. The tobacco industry has invested millions of dollars in research and advertising targeted to people of color, women, and low-income families [14]. They are the target of much of the $8 billion dollars tobacco companies spend on marketing each year. The tobacco industry also spent over $7 billion dollars in 2014 on discounts and coupons to keep the price point of cigarettes low, and to entice low-income smokers. Low-income communities have higher concentrations of tobacco retailers, and are more likely to have tobacco retailers near schools [4].

Societal and policy factors also make smoking cessation more difficult for some smokers. Nicotine has been found to be nearly as addictive as heroin and cocaine [15,16,17]. Thus, a person addicted to nicotine and addicted to smoking is rarely successful in stopping “cold turkey”; they need support, counseling, and, oftentimes, medications for successful quit attempts and prevention of relapse. Although there are undoubtedly a number of smokers who have no desire to quit smoking, it is estimated that 70% of smokers want to quit smoking [18]. Smokers from lower socioeconomic groups are as likely to attempt to quit smoking as those from higher socioeconomic groups, but are less likely to succeed [19,20]. Lack of health insurance coverage may pose challenges for lower SES smokers seeking cessation support, counseling, and medications [21]. Higher social capital, higher educational attainment, and stronger social networks are linked to an increased likelihood of smoking cessation [22,23,24], which may partially explain why members of marginalized groups are less likely to be successful in quit attempts.

Finally, a lack of agency and voice in the political process may also contribute to a dearth of laws and policies supporting members of marginalized groups. There is evidence to suggest that smokers are less likely to vote and to be involved in the political process than their non-smoking peers [25]. Solutions may be less forthcoming for marginalized groups because they may be more disconnected from the political process and, without an organized advocacy presence, have little ability to draw funding and resources to support tobacco cessation funding and policy.

## 2. E-Cigarettes, Youth, and SES

Although a comprehensive review of electronic nicotine delivery systems, or “e-cigarettes,” is beyond the scope of this article, it is important to note that e-cigarettes are changing the landscape of tobacco use. Usage patterns of e-cigarettes in vulnerable populations are not well described yet, but we do know that e-cigarette use, especially among youth, has increased exponentially. In 2018, over 3.6 million US middle and high school students reported using e-cigarettes within the past 30 days, which accounts for 4.9% of middle school students, and 20.8% of high school students in the United States [26]. The epidemic of youth e-cigarette use in the United States is concerning on many fronts. Exposure of the developing teenage brain to nicotine, a highly addictive substance, may negatively impact brain development [27]. E-cigarettes may contain harmful toxins that have pulmonary, cardiovascular, and other impacts [26]. Furthermore, and of greatest concern, a 2017 meta-analysis, including nine studies with 17,000 youth, demonstrated that youth who use e-cigarettes are significantly more likely to progress to smoking combustible cigarettes than youth who never tried e-cigarettes [27], thus giving rise to a new generation addicted to smoking.

The demographics of e-cigarette users versus conventional cigarette users are not well characterized. A 2018 study in the United States found that adult dual users were more likely to be white and more educated than combustible-only users [28]. Studies looking at the socioeconomic status of youth e-cigarette users are mixed—earlier studies found that teens with more disposable income had higher e-cigarette use [29,30], but studies that are more recent seem to suggest that youth living in lower income families are more likely to be susceptible to using e-cigarettes [31].

Lopez’s four-stage model [32] of the smoking epidemic in developed countries may give us insight into the evolving trajectory of e-cigarette use in marginalized communities. The model describes the initial rise in the use of combustible cigarettes in men and in communities with more wealth, which tend to have more disposable income and are more amenable to innovation. The model describes the eventual dissemination to women and lower SES communities. This model cautions us to be vigilant about patterns of e-cigarette use; it may be that wealthier, more educated youth begin using e-cigarettes and, as products become more widespread and affordable, their uptake in lower SES groups begins to increase.

E-cigarettes pose a series of challenges to pediatric providers, given the attractiveness of these devices to youth, the highly effective nicotine delivery, and the lack of cessation resources for youth addicted to e-cigarettes. Additional research and continued vigilance are needed in this area.

## 3. The Burden of Smoking: Physical, Economic, and Social Health

The impact of smoking and of secondhand smoke exposure are well documented. Smoking affects nearly every organ in the body and causes more deaths in the United States every year than HIV, illegal drug use, alcohol use, car accidents, and firearm incidents combined. Smoking causes 90% of deaths from lung cancer, 80% of deaths from chronic obstructive pulmonary disease (COPD), and increases the risk of death from all causes among both men and women [33]. The impact of secondhand smoke is similarly well documented. SHSE is associated with significant morbidity and mortality, including sudden infant death syndrome, pneumonia, ear infections, asthma, attention deficit disorder, and stroke. Adults exposed to SHSE are more likely to have heart disease, lung cancer, chronic lung disease, and stroke [34].

In addition to the compelling data on the health implications of smoking, there is a clear negative economic impact from smoking. A number of studies demonstrate the link between living in poverty and smoking [11,35,36,37,38]. A family in which a caregiver is addicted to cigarettes diverts funding to purchasing cigarettes which, depending on where the family lives, can cost nearly $400 a month for a pack-a-day smoker [39]. The redirection of critical disposable income may be at the expense of food, rent, clothing, or educational opportunities [40,41,42]. Additionally, smokers are less likely to be hired for jobs than non-smokers and, once hired, are paid less than non-smokers [43,44]. The lack of employment opportunities only serves to perpetuate the cycle of poverty in smokers. In the majority of US states, smokers can be charged higher premiums for health insurance, which can impede access to quit assistance, as well as to care for smoking-related illnesses [21]. Lack of appropriate medical care may lead to missed work and can add to familial economic instability. Poverty itself is significantly associated with smoking, even when adjusting for confounders, including mental health, nicotine dependence, and substance use [36]. Thus, tobacco use reinforces the cycle of poverty.

Finally, there are a number of studies suggesting that smokers experience more social isolation and have lower social capital than non-smokers [45,46,47]. There is a bidirectional, and possibly causal, relationship between teenagers smoking and possessing a negative affect compared with their nonsmoking peers [48], which may further contribute to social isolation. Low social capital and loneliness are linked to poor mental health and poor physical health [49]. Poor caregiver mental health has important implications on the health of children [50], further adding to the burden experienced by children whose parents smoke. Children of smokers who are more socially isolated may lack access to socially protective factors, which is thought to foster resilience [51].

## 4. Where Do We Go from Here? Clinical and Policy Interventions for Pediatric Providers

In order to meaningfully address smoking among marginalized populations, pediatricians should focus on clinical interventions, and on supporting policies that address family poverty, healthcare, and employment [35].

While adult physicians are screening for tobacco use and offering cessation assistance, there is room for improvement in the pediatric setting. Research in pediatrics shows that the vast majority of parents feel that it is a pediatrician’s job to ask about SHSE and that they would accept tobacco cessation assistance from the pediatrician, demonstrating that healthcare providers have a valuable role in actively supporting family members who want to quit smoking [52]. However, in already busy pediatric practices, pediatricians can feel uncertain of how to counsel parents on smoking cessation or be unaware of resources for parents to quit [53,54]. Fortunately, there are well-researched techniques pediatricians can use to help parents quit smoking, which are detailed in Table 3.

Pediatricians and others who provide care to children have a unique opportunity to serve as a resource for smokers seeking to quit. While adults (especially younger parents) may see their healthcare providers infrequently [55,56], pediatricians have a unique opportunity to offer quit assistance to families, as children see their pediatric provider 10 times before they reach two years of age [57]. Pediatricians can use the pediatric visit as an opportunity to offer extended family healthcare services. The 2015 American Academy of Pediatrics (AAP) Clinical Practice Policy has a number of potential action steps for providers. These include screening all children for tobacco use and tobacco exposure, including tobacco use prevention as part of anticipatory guidance, treating adult caregivers who use tobacco through providing connections to quit resources, and prescribing Food and Drug Administration (FDA)-approved tobacco cessation medications [58]. The CEASE (Clinical Effort Against Secondhand Smoke Exposure) study demonstrated that a simple framework for screening outpatient children for secondhand smoke exposure and helping parents trying to quit smoking was feasible and acceptable to parents and physicians [59,60].

Pediatric clinicians have an important role in counseling youth against starting to use tobacco. Ninety percent of adults addicted to nicotine started smoking before age 19 [66]. E-cigarettes are especially attractive to youth, as discussed above. Therefore, pediatricians should remain vigilant and counsel both parents and patients around the risks of tobacco use to prevent nicotine addiction from beginning.

The AAP also put forward the Public Policy to Protect Children from Tobacco, Nicotine, and Tobacco Smoke in 2015 with policy recommendations to protect children from tobacco smoke [67]. The uptake of these policies varies from state to state, but they include FDA regulation of all tobacco products, funding tobacco control efforts, prohibiting tobacco advertising and promotion aimed at children, prohibiting point of sale tobacco advertising and product placement that can be viewed by children, restricting the depiction of tobacco use in movies, banning the promotion and sales of e-cigarettes to youth, targeting interventions to marginalized groups who bear a greater burden of tobacco dependence, increasing tobacco prices, raising the purchase age of tobacco to 21, prohibiting flavoring, enacting comprehensive smoking bans, and prohibiting smoking in multiunit housing. In addition, the AAP recommends prohibiting children under 18 from working on tobacco farms and in tobacco production, as well as ensuring all nicotine liquids have child resistant packaging.

Policies such as universal health coverage will also ensure that smokers have the resources they need to support quit attempts, including both behavioral counseling as well as evidence-based cessation aid medications. Given the considerable economic burden placed on insurers of smokers such as employers, insurance companies, and Medicare/Medicaid administrators, they could also be targets for pediatricians to lobby for policy initiatives in tobacco cessation and abuse prevention. Poverty alleviation policies, such as universal basic income, can help to alleviate poverty, and programs to decrease barriers to higher education can help those living in poverty to improve their future earning potential.

## 5. Conclusions

Although the burgeoning use of e-cigarettes is changing the landscape and patterns of tobacco use, overall, combustible tobacco use trends in the United States are heading in the right direction. Nevertheless, many of the positive gains have been seen in people of high SES, with poorer families bearing a disproportionate burden of the total cigarette use, health/economic impacts, and attention from tobacco companies. To achieve equity in this area, the pediatric community must put additional energy into targeted research, policy, and cessation treatment for the vulnerable, yet resilient, populations at lower SES. Using innovative family-based interventions, pediatricians can play a vital role in the campaign to end the draining cycle of tobacco addiction.

## Figures and Tables

**Table 1 children-06-00031-t001:** Cigarette Smoking in Selected Populations.

Characteristics	% Smokers	Comment
**Income**		
Annual household income < $20K	32.2	As household income decreases, cigarette smoking increases.
Annual household income > $100K	12.15	
**Gender**		
Male	17.5	Men are more likely to smoke cigarettes than women.
Female	13.5	
**Ethnicity**		
American Indian/Alaska Natives	31.8	Cigarette smoking is highest among American Indians/Alaska Natives.
White	16.6	
Black	16.6	
**Education Level**		
GED (general education development certificate)	40.6	GED holders are nine times more likely to smoke than those with graduate degrees.
No diploma	24.1	
High school diploma	19.7	
Undergraduate Degree	7.7	
Graduate Degree	4.5	
**Mental Illness/Serious Pschological Distress**		
With serious psychological distress	35.8	Adults who had experienced serious psychological distress are more likely to be smokers.
Without serious psychological distress	14.7	

**Table 2 children-06-00031-t002:** Cigarette Smoking in Single Parents.

Characteristics of Single Parents	% Smokers	Comment
**Income**		
Below poverty level	34.4	Cigarette smoking increases with decreased household income.
At or above poverty level	27.7	
**Gender**		
Male	31.3	Fathers are more likely to smoke cigarettes than mothers.
Female	28.7	
**Ethnicity**		
White	39.0	By race, white single parents are the most likely to smoke.
Black	24.4	
Hispanic	15.2	
**Education Level**		
Some high school or less	34.6	Single parents with less education are more likely to smoke.
High school diploma/GED (general education development certificate)	34.2	
Bachelors or higher	14.8	
**WELFARE/TANF (Temporary Assistance for Needy Families) RECEIPT**		
Received income from welfare/TANF	44.8	Single parents who received assistance from TANF are more likely to smoke.
Did not receive income from welfare/TANF	28.6	
**Food Stamps/Snap (Supplemental Nutrition Assistance Program) Receipt**		
Authorized to receive food stamps/SNAP	38.6	Single parents who received assistance from SNAP are more likely to smoke.
Not authorized to receive food stamps/SNAP	22.8	

**Table 3 children-06-00031-t003:** Suggestions for Pediatricians to Help Parents/Caregivers Quit Smoking.

Practical Tips for Pediatricians to Help Patients’ Parents Quit Smoking	Comments	Evidence
**Ask**		
Treat smoke exposure as a vital sign to be measured at every visit.	Keep the screening question broad: “Does the child live or spend time with anyone who smokes inside or outside?”	Be aware that caregiver answers are typically accurate around 75% of the time [61].
**Assist**		
Use motivational interviewing (MI) techniques to encourage alliance and self-reflection with parents.	Even brief MI sessions increase smoking cessation attempts.	Using MI produces a 45% greater likelihood of quitting smoking compared to the control group [62].
Prescribe nicotine replacement therapy (NRT) with patches and gum to aid in quitting.	Most insurance plans pay for NRT when prescribed by a physician.	NRT can double the odds of achieving smoking abstinence [63].
**Refer**		
Connect parents to smoking quitlines able to do additional, longitudinal smoking counseling.	Many smoking quitlines help callers trying to stop vaping as well.	Counseling via a helpline can double the odds of long-term smoking cessation [64].
Consider also recommending texting programs or cessation phone apps to help parents stay engaged and accountable.	Think about signing up for a program to get a sense of it to better be able to recommend it to patients.	Smokers who received text messages to help quit were 38% more likely to abstain from smoking [65].

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
