# Peer review of "Tobacco Use as a Health Disparity: What Can Pediatric Clinicians Do?"

_children, 2019, doi:10.3390/children6020031_

Round 1

Reviewer 1 Report

This paper argues that tobacco use is a health disparity for children and suggests that addressing it may reduce the poverty cycle.  While the information presented does indicate the impact of this health behavior on children, both health-wise and economically, the discussion about pediatricians taking up a role in tobacco cessation with parents does not seem to add anything that the AAP has not already stated. A discussion about how this can be successfully embedded into a busy pediatric practice would have added a significant contribution to this area.

Additional comments:

line

24 update to current rate

31 Medicaid in not uninsured but rather federally insured for low income

32 spell out 75%

39 reference these reasons of state as suggested

68 don't need SHSE abbrev.

80 more current prices are available (including 2019)

86 cite

89-92 could be argued that equally true for any SES but perhaps low income would be less able to afford the help?

119-123 this seems to be a failure of the healthcare provider as those with Medicaid may have coverage for this (not necessarily so for all private insurance); see Simoneau, Hollenbach, et al 2017 - providers don't feel it is a priority among other ambulatory care needs

Reviewer 2 Report

The area of tobacco-related disparities is an important and relevant issue, particularly with respect to the role for clinical providers.  However, this paper could be improved by focusing its message to the clinical audience.  As such, the title needs to be adjusted to reflect the improved focus.  Tobacco as a health disparity is not a new concept, and efforts should be made to focus and identify specific roles for the practitioner.

Perhaps most importantly, the citations are often outdated and don't include one of the most comprehensive documents, National Cancer Institute's Monograph 22: A Socioecological Approach to Addressing Tobacco-Related Health Disparities. 

Lastly, there's no mention of electronic nicotine delivery systems (ENDS) which has reached significantly high prevalence rates among youth.  This emerging product presents new challenges for practitioners given its effective nicotine delivery and the lack of sufficient youth nicotine cessation resources. 

Reviewer 3 Report

Review article: “Tobacco Use as a Health Disparity”

The review raises several important questions that sparkles discussion regarding tobacco use and health disparity and deserves to be published; the purpose of the review is useful for the society. However, the review has several weaknesses that need to be addressed.

        I.            The abstract and the whole review is too short

     II.            Page1 Line 23 and 24: Author mentioned that increase in smoking cessation and decrees in smoking rate among US adults (fallen from over 40% in the 1960s to 16.8% in 2014). The readers may understand, this may be due smoking cessation but the past decade, there has been an increasing in use of electronic cigarette among the US youths and adults justify?

   III.            Include the table for smoker of low-income or high income, male female ratio, smokers of  low income families from the previous studied from different publication with references;   this may helpful for the readers

  IV.            Include the details of how does the secondhand smoke exposure affects the infants or children’s and also discuss about how does the children’s attracted towards tobacco products or e cigs in the form of crayon’s, chocolate                          

Round 2

Reviewer 1 Report

This is a vastly improved manuscript with more background, addition of vaping and real recommendations on what to do in clinical practice.

Reviewer 2 Report

The edits are sufficient for publication.

Children EISSN 2227-9067 Published by MDPI AG, Basel, Switzerland RSS E-Mail Table of Contents Alert
Back to Top